# Anti-Inflammatory and Antioxidant Activity of Hydroxytyrosol and 3,4-Dihydroxyphenyglycol Purified from Table Olive Effluents

**DOI:** 10.3390/foods10020227

**Published:** 2021-01-22

**Authors:** África Fernández-Prior, Alejandra Bermúdez-Oria, María del Carmen Millán-Linares, Juan Fernández-Bolaños, Juan Antonio Espejo-Calvo, Guillermo Rodríguez-Gutiérrez

**Affiliations:** 1Instituto de la Grasa, Consejo Superior de Investigaciones Científicas (CSIC), Campus Universitario Pablo de Olavide, Edificio 46, Ctra. de Utrera, km. 1, 41013 Seville, Spain; mafprior@ig.csic.es (Á.F.-P.); aleberori@ig.csic.es (A.B.-O.); mcmillan@ig.csic.es (M.d.C.M.-L.); j.fb.g@csic.es (J.F.-B.); 2Tecnofood ID Soluciones S.L. C/Paz nº 4, 3º G, 18.360 Huétor Tájar, 18016 Granada, Spain; jaespejo@hotmail.com

**Keywords:** table olive, phenolics, hydroxytyrosol, 3,4-dihydroxyphenylglycol, anti-inflammatory, antioxidant

## Abstract

New liquid effluents based on the use of acetic acid in the table olive industry make it easier to extract bioactive compounds to be used for food, cosmetic, and pharmaceutical purposes. The use of water acidified with acetic acid or in brine with or without acetic acid for storing the table olive enhances the extraction of two more active phenolic compounds: hydroxytyrosol (HT) and 3,4-dihydroxyphenylglycol (DHPG). This work has two aims: (1) measure the solubilization of phenolics controlled for two years using more than thirty olive varieties with different ripeness index as a potential source of HT and DHPG, and (2) evaluate the anti-inflammatory activity of the purified phenolics. The effluents with a higher concentration of phenolics were used for the extraction of HT and DHPG in order to evaluate its antioxidant and anti-inflammatory activity in vitro by the determination of pro-inflammatory cytokines such as Human Tumor Necrosis Factor-α (TNF), Interleukin-6 (IL-6), and Interleukin-1β (Il-1β). The anti-inflammatory activity of these phenolic extracts was demonstrated by studying the expression of cytokines by qPCR and the levels of these proteins by enzyme-linked immunosorbent assay (ELISA).

## 1. Introduction

The biological activities of olive phenols such as oleuropein, verbascoside, hydroxytyrosol (HT), tirosol (Ty), luteolin, and apigenin 7-O-glycosides, as well as phenolic acids, which are minor constituents of olive fruits, have been widely studied [1,2,3,4,5]. Among them, HT stands out because of its remarkable biological activities [6]. There are many methods of synthesis of HT for obtaining and purifying it from a natural procedure (olive oil by-product) [7], some of which are currently in place in the food industry [6,8]. Only recently was a similar phenol obtained and detected in some olive sources with a high potential as an antioxidant and anti-inflammatory agent, 3,4-dyhydroxyphenilglycol (DHPG) [9]; therefore, not much is known about its activity or its origin. Previous studies show its bioactive potential, such as its antioxidant character in different matrices such as water, pectin, or edible films [10,11,12] and overall, its synergistic effect was found with HT regarding the antioxidant activity in a hydrophilic matrix [13], the antioxidant effect in edible oils [9], and the inhibition of platelet aggregation [14]. The presence of phenols in virgin olive oil improves the stability of the fatty acid composition in addition to contributing to its beneficial properties [15]. DHPG is the principal metabolite of norepinephrine (NE), which is a sympathetic neurotransmitter that plays a critical role in regulating physiological processes related to the sympathetic activity in healthy and diseased states [16]. Thus, the high synergism found between HT and DHPG together with the presence of DHPG in the sympathetic activity make the further study of DHPG as well as the search for new extraction sources crucial. 

DHPG was found in olive oil wastes, mainly in alperujo or two-phase olive oil extraction waste, but the concentration was very low, mainly after the thermal treatment required for the solubilization of phenols into a liquid phase, and their extraction and purification were difficult [17]. On other hand, in a previous study, the presence of high concentrations of DHPG in natural olives has been reported [18], where low values in the ratio HT/DHPG indicate a good presence of this phenolic compound in table olives. The levels of DHPG found in all samples of natural black olives, regardless of cultivar type or processing method, were higher than in natural turning color or natural green olives. DHPG is present in natural brines during fermentation and storage, but it has not been detected in table olive treated with sodium hydroxide. The table olive is one of the most commonly consumed and popular snacks in the Mediterranean coastal zone. It contains a large range of polyphenols to which a multitude of beneficial properties is attributed, and it prevents chronic diseases such as cardiovascular, neurodegenerative, inflammatory disorders, or cancer [19]. The effluents of the olive table industry, mainly those untreated with sodium hydroxide, contain a great proportion of phenols and are a potential source of DHPG, which allows for the study of one of the most interesting properties of it. It is important to analyze the type of source used not only by the main phenols but also by their possible companions once they have been purified and also for the fact of making their extraction easier, avoiding undesired reactions. The intake of anti-inflammatory drugs is widespread, and they are used not only for their anti-inflammatory, analgesic, and antipyretic properties but also for their important beneficial effects such as cardiovascular protection and cancer prevention [20]. In this sense, the number of scientific studies to determine the anti-inflammatory effects of natural compounds has increased in the last few years with the aim of preventing a wide range of diseases. Thus, the extraction of natural phytochemicals from natural sources such as by-products can help to enhance not only human health but also environmental health. In this industry, a large amount of liquid effluents is generated, mainly from the fermentation and storage steps, where the brine is the most common medium used industrially [21]. Recently, changes in the table olive industry have led to replacing the high concentration of salt in brines by the use of acetic acid or even by storing them in water that has been acidified with acetic acid without sodium chloride [22]. Thus, the presence of phenolic compounds in these kinds of liquids could be a great opportunity to obtain industrially remarkable phenolics such as HT and DHPG. 

The aim of this work was to find new sources of by-products generated from the olive industry, which could contain DHPG in high concentration, and to purify it in order to evaluate its antioxidant and anti-inflammatory activity in vitro, which is one of its potential properties that has not been as extensively studied as the HT extracted from the same source. 

## 2. Materials and Methods

### 2.1. Materials

Samples from nine olive varieties with a ripeness index (RI) higher than 4 and twenty-seven olive varieties with an RI of between 1.5 and 2.5 were supplied by Tecnofood I+D Solution Company, Massanassa, Valencia, Spain, and these are shown in (Table 1). The determination of ripeness index of olive samples was carried out following the method described by García et al. (1996) [23], which is a well-used method based on the color of the skin of 100 olives divided into 8 groups being subjectively evaluated according to the following characteristics: group 0, skin bright green; group 1, skin green-yellowish; group 2, skin green with reddish spots; group 3, skin reddish-brown; group 4, skin black with white flesh; group 5, skin black with <50% purple flesh; group 6, skin black with 50% purple flesh; and group 7, skin black with 100% purple flesh. The ripeness index is determined by the equation where *i* is the number of the group and *ni* is the number of olives in it. The evaluation was performed in triplicate.
ripeness index = ∑(*ini*)/100(1)

### 2.2. Chemicals

Sodium chloride (NaCl) has been used as conservation liquid; HPLC grade acetonitrile, fluorescein, Trolox, and 2,2′-azobis (2-amidine-propane) dihydrochloride (AAPH) were obtained by Sigma-Aldrich (Madrid, Spain). Ultrapure water was obtained from a Milli-Q water system (Millipore, Milford, MA, USA). The following reagents were used for cell growth: Medium RPMI (Roswell Park Memorial Institute, Buffalo, NY, USA) 1640 w/l-Glutamine w/25 mM HEPES (4-(2-hydroxyethyl)-1-piperazineethanesulfonic acid), Fetal Bovine Serum (FBS), Trypsin 0.25% in PBS w/o Calcium w/o Magnesium w/Phenol red and Phosphate-buffered saline w/o Calcium w/o Magnesium (PBS) were purchased from Biowest (Riverside, CA, USA). A penicillin/streptomycin (P/S) solution was obtained from Gibco^®^ (Life Technologies SA, Alcobendas, Spain). Thiazolyl blue tetrazolium bromide Biochemical (MTT), dimethyl sulfoxide Cell culture grade (DMSO), and Trypan Blue 0.4% solution in PBS were obtained from Applichem (Darmstadt, Germany). 

Phorbol 12-myristate 13-acetate (PMA), lipopolysaccharide from *Escherichia coli* O55:B5 (LPS), and TritonTM X-100 came from Sigma Chemical Co. (St. Louise, MO, USA). TRIsure^TM^ was purchased from Bioline (London, UK). For performing reverse transcription, an iScript^TM^ cDNA Synthesis Kit was used, and for Real-time Quantitative PCR, a SYBR Green Master Mix was used; both were from Bio-Rad Laboratories (Hercules, CA, USA). Primers were purchased from Eurofins Biolab S.L.U. (Barcelona, Spain). Human Tumor Necrosis Factor-α (TNF-α), Interleukin-6 (IL-6), Interleukin-1β (IL-1β), and Interleukin-10 (IL-10) enzyme-linked immunosorbent assay (ELISA) sets were obtained from Diaclone Biotech (Besançon, France).

### 2.3. Isolation of Hydroxytyrosol (HT) and 3,4-Dihydroxyphenylglycol (DHPG) from Olive Oil By-Products

The liquid phase, with higher contents of HT and DHPG, was used for the isolation of these two phenols. The methods to purify HT and DHPG have been described and patented by Fernández-Bolaños et al. (2004) [24] and Fernández-Bolaños et al. (2010) [25], respectively. Both methods are based on physical chromatographic systems that allow the extraction of natural compounds without any organic solvent or chemical or enzymatic reactions, obtaining a purity degree over 95%, referring to dry matter. 

### 2.4. Study of HT, Ty, and DHPG in Brines and Acid Solutions

In order to study the solubilization of HT, DHPG, and Ty during the storage of different olive varieties, a total of 36 varieties of natural turning color olives were collected from four different regions of Spain (Cadiz, Cordoba, Seville, and Granada) and put in three different conservation liquids for 24 months, with an average range of 0.7 L per 400 g of olives. Duplicate analyses were performed on each sample. The samples were placed in three preservation liquids based on two state of maturity (green and ripe). The conservation liquids used were those as follows: NaCl at 10%; NaCl 6–8%; and 1% acetic acid (pH ˂ 4.2); 1% acetic acid. The samples were checked 8 times in 24 months, and the amounts of DHPG, HT, and Ty were determined by HPLC-DAD.

### 2.5. HPLC-DAD Analysis

The phenolic profile was determined using a high-resolution liquid chromatography system, HPLC (Hewlett-Packard 1100 series equipped with an array diode detector and an Agilent 1100 series automatic injector which introduces 20 µL of sample). The chromatographic column used was Teknokroma Tracer Extrasil OSD2 of 5 µm particle size and dimensions of 25 × 0.46 internal diameter. As an eluent, HPLC grade acetonitrile (B) and mill-Q water with 0.01% in Trifluoroacetic acid (TFA) were used. The flow rate was 1 mL/min and the chromatograms were recorded at 254, 280, and 340 nm. Phenolic compounds were separated using the following gradient: 0–30 min, 5% B; 30–45 min, 25% B; 45–47 min, 50% B; 47–50 min, 0% B.

The identification and quantification of phenolic compounds were based on the comparison of the retention times (RT) and absorbance values of detected peaks in solvents with those obtained by the injection of pure standards of each analysis.

### 2.6. Antioxidant Activity Assays: Oxygen Radical Absorbance Capacity (ORAC) Assay

The inhibition of oxidation induced by peroxyl radicals produced by thermal oxidation of AAPH in a sample is known as the oxygen radical absorbance capacity (ORAC) assay, and it was performed following Bermúdez-Oria et al. (2019) [12]. The reactive oxygen species (ROS) produced diminishes the fluorescence signal generated by the fluorescein. The samples were conveniently diluted with sodium phosphate buffer (10 mM, pH 7.4), and 25 µL of the sample was transferred to a microplate. A blank with 25 µL of phosphate buffer was used, while the standards were filled with 25 µL of Trolox solutions at concentrations between 10 and 140 µM. Subsequently, 150 µL of 1 µM fluorescein was added to all wells. The plate was incubated at 37 °C for 15 min. After this time, 25 µL of AAPH (250 mM) were added to each well to initiate the reaction. In a plate reader (Fluoroskan Ascent™, Thermo Scientific™, Waltham, MA, USA), measurements were taken every 5 min for a period of 90 min at an excitation wavelength of 485 nm and an emission wavelength of 538 nm. Final ORAC values were calculated using the regression equation between Trolox concentration and area under the curve (AUC), and they were expressed as µmoles of Trolox equivalents per mmoles of active molecules.

### 2.7. Cell Culture and Treatments

The human monocytic THP-1 cell line was graciously provided by the Cell Biology Unit (Institute of Fats, CSIC, Seville, Spain), and it was cultured in suspension with RPMI 1640 medium using a Thermo Forma Series II Water Jacketed CO_2_ Incubator by Thermo Fisher Scientific at 37 °C and 5% CO_2_. The medium was supplemented with 1% (P/S) and 10% heat-inactivated FBS for 30 min at 56 °C for the purpose of inactivating proteins that could interfere in the immune response. Monocytes were differentiated from macrophages by incubation with 100 nmol/L of PMA for 4 days. Adherent macrophages were treated with serum-free RPMI 1640 for 24 h. Cell work was carried out in a Telster^TM^ Bio II Advance 4 Biological Safety laminar flow cabinet, and cells were examined under a Culture Microscope model Olympus CK40-CPG (London, UK).

### 2.8. Citotoxicity Assay 

Once grown, the cells were seeded at 50,000 cells per well in 96-well plates and incubated at 37 °C and 5% CO_2_ for 24 h. The concentrations used were as follows: 10, 40, 60, 80, and 100 μg/mL. Afterwards, 100 μL of MTT (5 mg/mL) were added and incubated for 3 h, thus favoring the formation of formazan crystals. Finally, the MTT was removed, and 100 μL of dimethyl sulfoxide were added to dissolve the formazan. Triton^TM^ X-100 was used as a positive control, and cells in the medium were used as a negative control. Absorbance was measured in a Thermo Scientific Multiskan Spectrum spectrophotometer with a microplate reader by Thermo Fisher Scientific at 570 nm. 

### 2.9. Cell Treatment

The human monocytic THP-1 cell line was cultured in suspension in RPMI 1640 supplemented with 10% heat-inactivated FBS and 1% P/S. Cells were seeded at 5 × 10^5^ cells/well in 12-well plates and differentiated to macrophage-like cells by treating for 4 days with PMA at 100 nmol/L [26]. In order to induce inflammatory damage to incubated cells, LPS was added in a final concentration of 0.05 μg/mL; then, the cells were incubated for 1 h. After that HT, DHPG and a mixture of them were added to 50 μg/mL and incubated for 24 h. 

### 2.10. RNA Extraction, RT PCR, and Quantitative PCR

After LPS stimulation, the total RNA of adherent cells was extracted using TRIsure^TM^ Bioline (London, UK). The evaluation of RNA concentration was made with spectrophotometry NanoDrop ND-1000 Uv-Vis by Thermo Scientific at 260 and 280 nm. The concentration value was provided by the device. These measurements were conducted three times for each sample, and the RNA quality obtained had a purity degree of A260/A280. Momentarily, RNA (1 μg) was subjected to reverse transcription (iScript, Bio-Rad, Madrid, Spain) with iScript™ Reverse Transcriptase (Bio-Rad) according to the manufacturer’s protocol using a MJMini^TM^ Personal Thermal Cycler by Bio-Rad. In order to perform real-time PCR amplification, 10 ng of the cDNA were taken as a template. The mRNA were quantified using iTaq Universal SYBR^®^ Green qPCR SuperMix (Bio-Rad) containing primer pairs for TNF-α, Il-6, Il-1β, and Il-10. The Thermal Cycler used was a CFX96 Connect Real-Time System and the iCycler data analysis software (Bio-Rad, Hercules, CA, USA). The amplified cDNA levels were compared among different groups using the standard 2^−(ΔΔCt)^ method. The reference gene hypoxanthine phosphoribosyltransferase 1 (Hprt) and glyceraldehyde 3-phosphate dehydrogenase (Gapdh) were used as “housekeeping”. All amplification reactions were tested in triplicate, and the sequence of oligonucleotides used was corresponding as follows: Tnf-α (NM_000594): 5′-TCCTTCAGACACCCTCAACC-3′ and 5′-AGGCCCCAGTTTGAATTCTT-3′; Il1-β (NM_000576): 5′-GGGCCTCAAGGAAAAGAATC-3′ and 5′-TTCTGCTTGAGAGGTGCTGA-3′; Il-6 (NM_000600): 5′-TACCCCCAGGAGAAGATTCC-3′ and 5′-TTTTCTGCCAGTGCCTCTTT-3′; Gapdh (NM_001289746): 5′-CACATGGCCTCCAAGGAGTAAG-3′ and 5′-CCAGCAGTGAGGGTCTCTCT-3’; Hprt 1(NM_000194): 5′-ACCCCACGAAGTGTTGGATA-3′ and 5′-AAGCAGATGGCCACAGAACT-3′.

### 2.11. Cytokine Determination by Enzyme-Linked Inmunosorbent Assay (ELISA)

The levels of TNF-α, Il-6, and IL-1β in culture were determined by enzyme-linked immunosorbent assay (ELISA) in 96-well microtiter MaxiSorp plates from Nunc using ELISA kits according to the manufacturer’s protocol. Using calibration curves from serial dilution of human recombinant standards in each assay, the concentrations of cytokine were estimated and expressed in pg/mL.

### 2.12. Stadistical Analysis

All values are expressed as arithmetic means ± standard deviations (SD). Data were evaluated with Graph Pad Prism Version 5.01 software (San Diego, CA, USA). The statistical significance of any difference in each parameter among the groups was evaluated by one-way analysis of variance (ANOVA), following the Tukey multiple comparison test as post hoc test. *p* values of less than 0.05 were considered statistically significant.

## 3. Results and Discussion

### 3.1. Study of HT, Ty, and DHPG in Brines and Acid Solutions

Nine samples of different varieties with a higher ripening index (RI > 4) were used for the trial with all three types of liquids: brines, brine plus acid, and acid. Figure 1 shows the evolution of the contents of the three main simple phenolics over a two-year period. The greatest increase in concentration was observed in the case of hydroxytyrosol, especially in some varieties and where brine concentrations above 350 mg/100 g of fresh olives were reached. The use of acid accompanied or not by brine favored the extraction of tyrosol after seven weeks of storage. In the case of DHPG, the trend was that the use of acid favored a slight increase in its concentration in the absence of brine, clearly highlighting the 8M sample in the three treatments, which reached values of 120 mg/100 g of fresh olives with the acid. It was also observed that in most of the samples, the concentrations in the three phenols decreases from the year of storage in a gradual way, which could be the result of an extractive decrease and an increase in the oxidative degradation of these compounds.

In the case of the greener olives (1.5 ≤ RI ≤ 2.5) (Figure 2), when the number of varieties was higher, a greater dispersion of data was observed, especially in the release of hydroxytyrosol and tyrosol, with the behavior of DHPG being more homogeneous. The maximum HT values were above 400 mg/100 g of fresh olives after five months of storage, while those of Ty were between 70 and 120 mg/100 g of fresh olives for a lower number of samples. The maximum value for DHPG was 140 mg/100 g of fresh olives for a sample preserved in brine after 12 months. In this last phenol, the concentration started to increase from the fifth month of storage, so this period was necessary for the recovery of DHPG. 

In order to see the behavior of each of the ripest and greenest varieties in terms of the solubilization of phenols more clearly, the concentrations in the two most important, HT and DHPG, are presented in (Figure 3). According to the need to recover one or the other or both, certain varieties offered important advantages. In the case of the more mature varieties, the 8M variety stood out for its high concentration in DHPG, which varied as a function of the medium between 80 and 120 mg/100 g of fresh olives, while maintaining a less variable range of HT between 270 and 300 mg/100 g of fresh olives. Other samples such as 2M presented a high HT concentration (370 mg/100 g of fresh olives) in brine and very low concentration in DHPG, halving its HT content in the case of using acid with or without brine. If a source is sought to obtain a mixture of HT with DHPG to take advantage of the synergistic effect of both compounds and improve their biological activity, all samples are useful, mainly the 8M. However, in the case where only DHPG is sought for use as a phytoregulator in which the presence of HT must be avoided, samples such as 1M or 7M in any medium would be preferably suitable. 

The samples of the least ripe olives presented a different behavior; it seemed that most of the samples between 74% and 88% presented a concentration in DHPG below 40 mg/100 g of fresh olives for the three media used. In the case of the brine where the highest percentage was between 0 and 20 mg/100 g of fresh olives (63%), the addition of acid slightly improved the concentration in such phenol, but it was always below 40 mg/100 g of fresh olives in most samples. The 16V sample stood out for its high DHPG content, which reached over 140 ppm in brine, and it was between 300 and 400 mg/100 g HT from fresh olives in all media. As for HT solubilization, there was a great variation, depending on the type of olive. 

In general, the ripest varieties showed a better solubilization of DHPG, which was not so clear in the case of HT. As a source of HT, most olives could be used in all three media; however, for DHPG, it seemed that the use of the ripest olives with the addition of acid was more effective.

To compare the results with those observed in the literature, the range of maximum concentrations expressed in µmol/L of storage liquid reached in the ripest olives was 335–4030 and 5190–13,715 for DHPG and HT, respectively. In the case of the greenest olives, they were 168–4869 and 1853–15,198 µmol/L for DHPG and HT, respectively. There has been hardly any work done to determine the amount of DHPG in table olives and their fermentation, storage, or packaging liquids. Most of the works determine the amount of HT and its derivatives as hydrolysis products of oleuropein as the main cause of bitterness. Industrially, to eliminate this bitterness, either prolonged storage is carried out or time is reduced through the use of an alkaline treatment with NaOH. This treatment rapidly hydrolyzes the oleuropein and degrades other phenols such as DHPG. In one of the studies [27] where the solubilization of phenols from soda-treated olives was analyzed, it was observed that the maximum concentration in HT after two months of storage was 3.5 mmol/L for the manzanilla variety, or 780 mg of HT/g of fresh olive, which is higher than the values reported in this work from natural olives because of the use of NaOH.

Since the degradation of DHPG takes place at high pH, sources of this phenolic in the table olive industry have to be through natural processing where NaOH is not used. One of the few studies carried out with naturally processed olives where it was possible to quantify DHPG and HT showed that these results are above those shown in the literature [18], where the concentrations in these phenols in the packing media for natural olives produced without NaOH were determined. In this study, the maximum values found in conservation brine for the Empeltre variety of black olives were 1810 µg/L DHPG and 7933 µg/L HT for the Manzanilla variety. These values are again much lower than those found in this study, which can be explained by the fact that the maximum reached in this work corresponded to storage periods that may have been different from those reported in the literature and that the varieties tested had been different.

### 3.2. Extract Preparation

Three extracts were prepared for the antioxidant and anti-inflammatory activity trials. One was rich in HT from the mixture of the 27V and 23V samples, the second was rich in DHPG from the 1M, 1V, 10V, and 15V samples, and the third was a mixture of HT and DHPG using a mixture of the 8M and 16V samples. The three kinds of solutions were also mixed to obtain sufficient amounts for each purification, which was made using only water through a chromatographic system as described in the patents mentioned and without any organic solvents.

### 3.3. Antioxidant Activities 

The antioxidant capacity of the three extracts was evaluated by their oxygen radical absorbance capacity (ORAC). The results obtained for the measurement of antioxidant activity by the ORAC of the sample containing a mixture of the two phenols in the ratio 1:1 and of the two phenols individually are shown in (Figure 4). It can be seen that the extract with the highest antioxidant capacity is the one rich in DHPG with 9758 μmol of Trolox/mmol of active molecule compared to 7042 μmol of Trolox/mmol of the active molecule of the mixed extract, which is well above the HT extract, which reached 4127 μmol of Trolox/mmol of active molecule. 

This same behavior was observed when these molecules were linked to soluble dietary fibers from the strawberry cell wall; these results were obtained after a simulated in vitro digestion with gastric and intestinal fluids. The same result was found in two different methods (DPPH (2,2-diphenyl-1-picrilhidrazilo) and ABTS (2,2′-azino-bis(3-ethylbenzothiazoline-6-sulfonic acid)) and suggests that the extra OH group present in the DHPG with respect to the HT molecule confers greater antioxidant capacity to the DHPG [13]. On the other hand, one could consider a possible synergistic effect when using a DHPG–HT extract in a 1:1 ratio where an increase in the antioxidant capacity is appreciated with respect to an extract solely composed of HT. 

### 3.4. Viability of Cells

A cell proliferation assay was performed by the metabolic reduction of 3-(4,5-dimethylthiazol-2-yl)-2,5-diphenyltetrazole bromide (MTT) in monocytes cells in order to determine the non-toxic effect of the extracts under study and, thus, to be able to evaluate their anti-inflammatory capacity. They were conveniently differentiated into adherent macrophages, which were taken as a model of human macrophages derived from THP-1 cell line, whose immortal cell model served to study the anti-inflammatory response against the use of macrophages of tissue origin. This type of test is one of the most commonly used for its simplicity and speed [28]. 

In monocytes derived from THP-1, the toxicity of the extracts tested in concentrations from 10 to 100 ppm with respect to the control of living cells after 24 h of treatment was very low or null; or in other words, the cellular viability was very high or 100%, and there were no significant differences with respect to the control, so it could be said that DHPG, HT, and the mixture of both extracts did not exert any damage on the cellular integrity in this cellular model. As shown in (Figure 5), the HT, DHPG extracts, and their mixture did not show any growth inhibition effects in human cells in a dose-dependent way. The capacity of HT to protect healthy cells has been reported in other studies where it was observed that HT presented a viability of more than 60%, even when they had undergone previous oxidation with H_2_O_2_ [29]. In fact, the anti-proliferative capacity of HT has been reported in other unhealthy cell lines such as HeLa and DG75 [30], which was very promising in the fight against neurodegenerative diseases.

### 3.5. Anti-Inflammatory Activity

One of the objectives of this work is focused on the verification of the anti-inflammatory biological properties of HT and DHPG extracts obtained individually and in combination.

In order to analyze the anti-inflammatory capacity in vitro, tests were carried out with LPS, which is a lipopolysaccharide from Escherichia coli and induces inflammation in the phagocytic cells that secrete pro-inflammatory cytokines (TNF-α, Il-6, and IL-1β). The release of these cytokines by the cells is the main indicator of the inflammatory state, considering that if the extract inhibits the gene expression of any pro-inflammatory cytokine, it could be used as a possible remedial drug to fight inflammation. 

Once the cell damage has been produced, the cells adhered to the culture plate are separated from the supernatants, and the gene expression study is carried out using quantitative PCR techniques for the three pro-inflammatory cytokines. In THP-1, the results obtained indicate that the three extracts tested reduced the gene expression of the cytokines TNF-α, Il-6, and IL-1β against the control of LPS (Figure 6). 

In the case of the tumor necrosis factor, the HT and DHPG extracts were significantly lower than the control, while in the mixture of the two phenols, this difference was not appreciated. In the case of Interleukin IL-1β, a drastic reduction was observed for the three extracts tested as well as in the case of Interleukin IL-6, where the DHPG extracts and the mixture showed an expression that did not present significant differences compared to the control of undamaged cells. There are studies that affirm that the HT isolated from virgin olive oil has anti-inflammatory activity in THP-1, inhibiting the antioxidant action by suppressing the induction of NO release by LPS in THP-1 in addition to the decrease in the gene expression of TNF-α, inducible nitric oxide synthase i-NOS, and cyclooxygenase COX-2 [31,32]. There are also authors who affirm that HT improves the ulcerative colitis in models of peritoneal cells in murine induced by dextran sodium sulfate (DSS) [33]. However, to our knowledge, there are no studies on the evaluation of pro-inflammatory cytokines that demonstrate the anti-inflammatory capacity as in the case of DHPG and HT–DHPG mixture, and the trials presented in this work demonstrate for the first time that these molecules could be used as possible inhibitors of inflammatory processes.

On the other hand, the levels of TNF-α, IL-1β, and IL-6 have been quantified in the cells’ supernatants through enzyme-linked immunosorbent assays (ELISA) for each cytokine, and the cells treated with DHPG were shown to secrete a smaller amount of cytokines TNF-α and IL-6 to the medium than the control with the inflammation inducer, which coincides with the results obtained from the gene expression. On the other hand, the cells treated with the HT extract decreased the concentration in the cytokine Il-6 in the medium with regard to the control. The synergic power of the DHPG and HT mixture in the secretion of IL-1β and IL-6 is significant, and it could add a new biological property of synergy to this phenolic union.

## 4. Conclusions

The effluents generated in the table olive industry during the natural fermentation and the storage in brine, brine plus acetic acid, or an acetic acid solution alone have been shown in the present study as important sources of HT and DHPG. The best results for the solubilization of the three phenols have been obtained when the acid has been used in the storage of the analyzed olives. The use of brine and acid separately or together leads to different solubilization of the three phenolics studies depending on the olive variety and the ripeness index, but all the cases produce liquid sources in which the phenolics can be industrially extracted. Despite the fact that the table olive industry is seasonal, storage in brine and/or acid improved the solubilization of phenolics for up to 12 months, which can be used throughout the year for the extraction of these bioactive compounds. The anti-inflammatory activity of these phenolic extracts was demonstrated by studying the expression of cytokines by qPCR and the levels of these proteins by ELISA. It can be through these results that it is possible to obtain phenolic compounds with anti-inflammatory capacity that could be used as good drugs to treat diseases of inflammatory origin.

## Figures and Tables

**Figure 1 foods-10-00227-f001:**
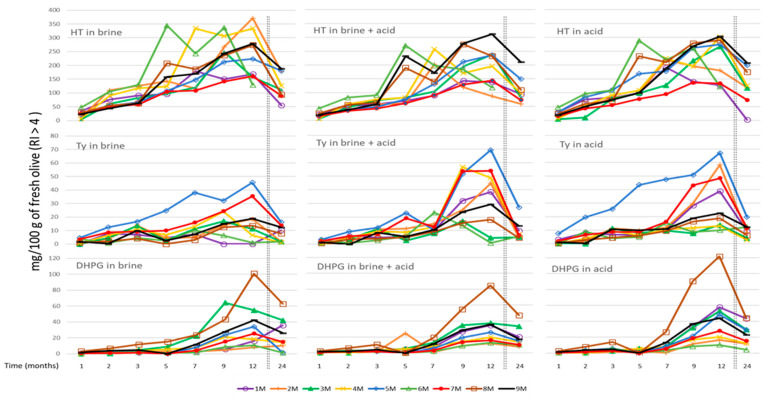
Solubilization of hydroxytyrosol (HT), tirosol (Ty) and 3,4-dihydroxyphenylglycol (DHPG) during 24 months in three different storage liquids (brine, brine in acid, and acid) for nine varieties of olives with a ripeness index higher than four.

**Figure 2 foods-10-00227-f002:**
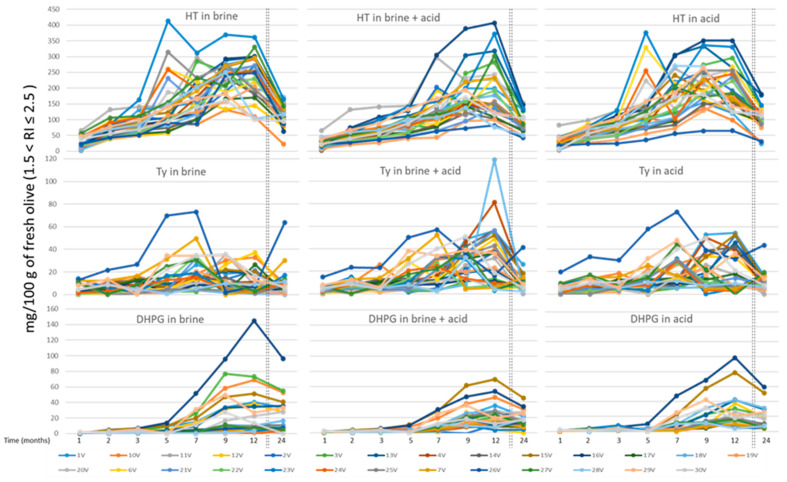
Solubilization of HT, Ty, and DHPG during 24 months in three different storage liquids (brine, brine in acid, and acid) for thirty varieties of olives with a ripeness index between 1.5 and 2.5.

**Figure 3 foods-10-00227-f003:**
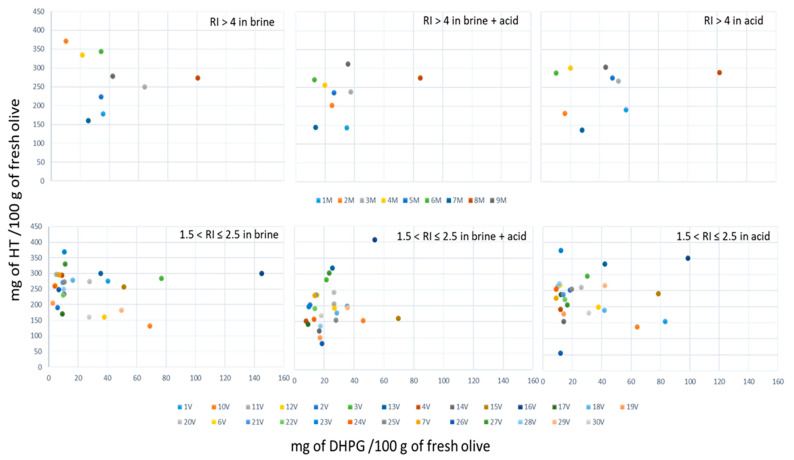
Maximum concentration of HT versus maximum concentration of DHPG achieved during the 12 months of storage for both ripeness stages of olives—higher than 4 (nine samples) and between 1.5 and 2.5 (thirty samples).

**Figure 4 foods-10-00227-f004:**
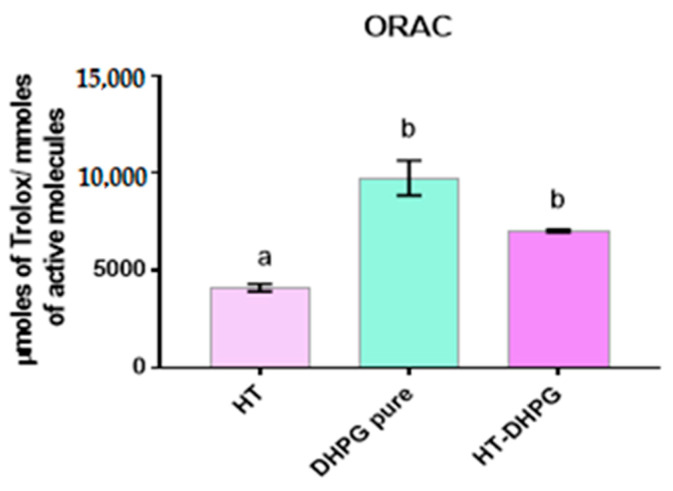
μmoles of Trolox equivalents obtained in each extract per mmoles of active molecule. Different letters indicate that there are significant differences between the results of the different extracts (*p* < 0.05).

**Figure 5 foods-10-00227-f005:**
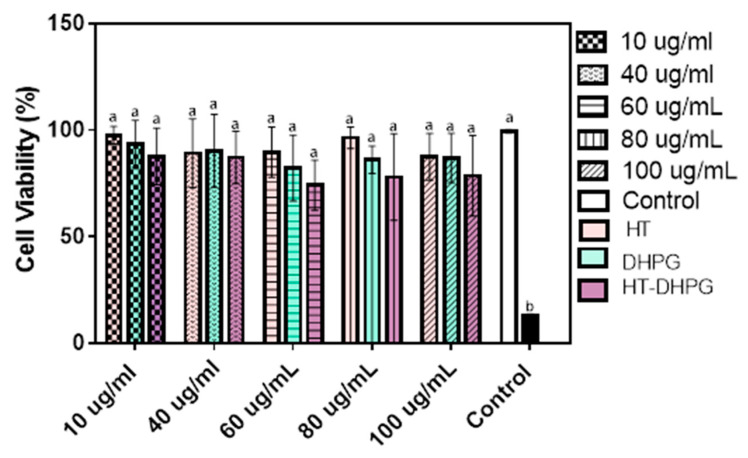
Cell viability (%) determined by 3-(4,5-dimethylthiazol-2-yl)-2,5-diphenyltetrazole bromide (MTT) assay, in the presence of three different extracts HT, DHPG, and HT–DHPG, after 3 h of treatment and different concentrations (10, 40, 60, 80, and 100 µg/mL). Values marked with different letter are significantly different (*p* < 0.05).

**Figure 6 foods-10-00227-f006:**
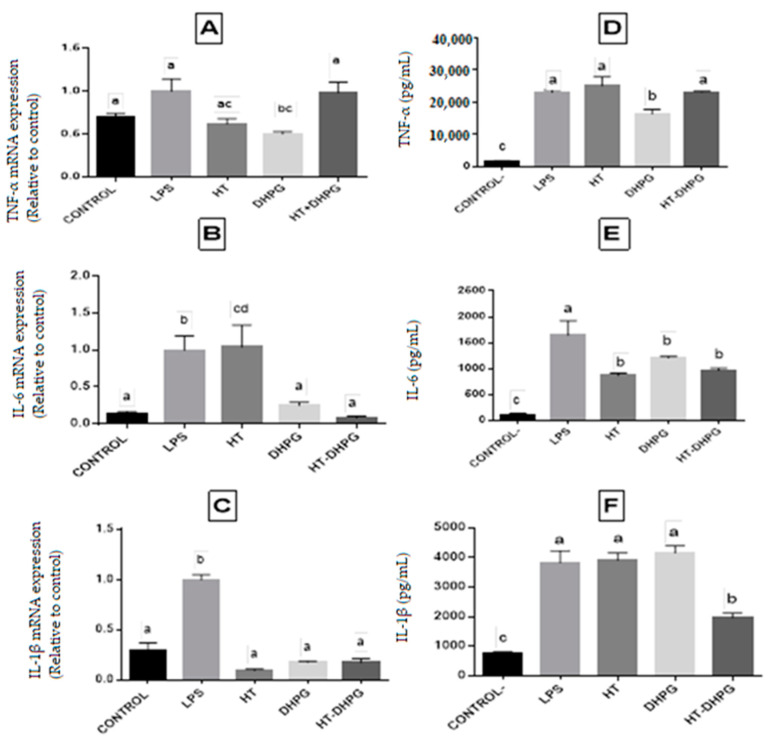
Human Tumor Necrosis Factor-α (TNF) (**A**), Interleukin-6 (IL-6) (**B**), Interleukin-1β (IL-1β) (**C**) mRNA expression in THP-1-derived macrophages after 24 h of treatment with or without lipopolysaccharide (LPS) (100 ng/mL) and HT, DHPG and HT–DHPG. Cytokine secretion of TNF (**D**), IL-6 (**E**), and IL-1β (**F**) production of THP-1-derived macrophages after 24 h of treatment with or without LPS (100 ng/mL) and HT, DHPG, and HT–DHPG. Values marked with different letter are significantly different (*p* < 0.05).

**Table 1 foods-10-00227-t001:** Samples of olives used for phenolic solubilization in brine, brine plus acid, and acid alone.

Codes	Denomiation	Origin	Date
Ripeness index (RI > 4)			
1M	Alameño	Alhama (Granada)	1/11/2016
2M	Hojiblanca	Alcalá del Valle (Cadiz)	18/11/2016
3M	EP	West of Granada	18/11/2016
4M	Rara 2	Montefrío (Granada)	18/11/2016
5M	Azul 1	Alhama (Granada)	17/12/2016
6M	Hojiblanca	Écija (Seville)	5/12/2016
7M	Azul 2	Alhama (Granada)	
8M	Loaime	Alhama (Granada)	8/12/2016
9M	Alameño	Alhama (Granada)	8/12/2016
Ripeness index (1.5 < RI ≤ 2.5)			
1V	Alameño	Alhama (Granada)	1/11/2016
10V	Nevadillo	Alhama (Granada)	1/11/2016
11V	Hojiblanca	Moriles (Córdoba)	
12V	Picual	Alcalá del Valle (Cadiz)	18/11/2016
2V	Hojiblanca	Alcalá del Valle (Cadiz)	18/11/2016
3V	EP	West of Granada	18/11/2016
13V	Lucio	West of Granada	
4V	Rara 2	Montefrío (Granada)	18/11/2016
14V	Gordal	Granada	18/11/2016
15V	Chorreo	Montefrío (Granada)	18/11/2016
16V	ISI-151	Montefrío (Granada)	18/11/2016
17V	COR-IG	Montefrío (Granada)	18/11/2016
18V	RAN-IG	Montefrío (Granada)	18/11/2016
19V	Picual	Écija (Seville)	5/12/2016
20V	Carrasqueño	Alcaudete (Jaén)	30/11/2016
6V	Hojiblanca	Écija (Seville)	5/12/2016
21V	Picual	Guadahortuna (Granada)	6/12/2016
22V	Lucio 80	Íllora (Granada)	13/12/2016
23V	Lucio 64	Íllora (Granada)	13/12/2016
24V	Alameño	Montilla (Cordoba)	13/12/2016
25V	Royal	Rabanales (Cordoba)	13/12/2016
7V	Azul 2	Alhama (Granada)	13/12/2016
26V	Rapasayo	Rabanales (Cordoba)	13/12/2016
27V	Ocal	Rabanales (Cordoba)	13/12/2016
28V	Hojiblanca	Montefrío (Granada)	23/12/2016
29V	480-1G	West of Granada	13/01/2017
30V	Brillante	Montefrío (Granada)	13/01/2016

## Data Availability

The data are available from the corresponding author upon reasonable request.

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
