# Peer review of "Anti-Inflammatory and Antioxidant Activity of Hydroxytyrosol and 3,4-Dihydroxyphenyglycol Purified from Table Olive Effluents"

_foods, 2021, doi:10.3390/foods10020227_

Round 1

Reviewer 1 Report

I have the following comments and suggestions:

  1. The aim is not written at the beginning of the abstract.
  2. The stability of phenols in olives and olive oil should be mentioned in the Introduction part, the following reference can be used:Dordevic, D., Kushkevych, I., Jancikova, S., Zeljkovic, S. C., Zdarsky, M., & Hodulova, L. (2020). Modeling the effect of heat treatment on fatty acid composition in home-made olive oil preparations. Open Life Sciences15(1), 606-618.
  3. Line 76: The aim should be written in the past tense.
  4. Line 80-84: this part is not belonging to the Introduction part.
  5. Line 87: how ripeness index was evaluated, it should be described.
  6. Line 128: Describe the exact mobile phase.
  7. Line 140: Describe in detail how ORAC assay was performed.
  8. Line 202: How the homogeneity of results was tested?
  9. Principal component analysis was not done, it should be performed. 

Author Response

Reviewer 1:

I have the following comments and suggestions:

    The aim is not written at the beginning of the abstract.

Response: The following sentence has been insert in the abstract to define the aim:The aim of this work was measure the solubilization of phenolics controlled for two years using more than thirty olive varieties with different ripeness index as a potential source of HT and DHPG, and the evaluation of the anti-inflammatory activity of the purified phenolics.

    The stability of phenols in olives and olive oil should be mentioned in the Introduction part, the following reference can be used:Dordevic, D., Kushkevych, I., Jancikova, S., Zeljkovic, S. C., Zdarsky, M., & Hodulova, L. (2020). Modeling the effect of heat treatment on fatty acid composition in home-made olive oil preparations. Open Life Sciences, 15(1), 606-618.

Response: the stability of phenols has been mentioned in the introduction and the reference has been also used: “The presence of phenols in virgin olive oil improve the stability of the fatty acid composition in addition to contributing to its beneficial properties [16]”.

    Line 76: The aim should be written in the past tense.

Response: It has been corrected.

    Line 80-84: this part is not belonging to the Introduction part.

Response: It has been deleted.

    Line 87: how ripeness index was evaluated, it should be described.

Response: It is already referenced LINE 90. The method for the determination of the ripeness of the olive samples is the one described by García et al., 1996. A description of this method has been insert in the text:

The determination of ripeness index of olive samples was carried out following the method described by García et al. It is a method where the color of the skin of 100 olives divided into 8 groups is subjectively evaluated according to the following characteristics: group 0, skin bright green; group 1, skin greenyellowish;group 2, skin green with reddish spots; group 3, skinreddish-brown; group 4, skin black with white flesh; group 5,skin black with <50% purple flesh; group 6, skin black with50% purple flesh; and group 7, skin black with 100% purple flesh. The ripeness index is determined by the equation where i is the number of the group and ni the number of olives in it. The evaluation was performed in triplicate.

ripeness index =å(ini)/100

    Line 128: Describe the exact mobile phase.

Response: This is already described: As eluent, HPLC grade acetonitrile and mill-Q water with 0.01% in TFA were used.

    Line 140: Describe in detail how ORAC assay was performed.

Response: The following description has been insert in the text:

The inhibition of oxidation induced by peroxyl radicals produced by thermal oxidation of 2,2′-azobis (2-amidine-propane) dihydrochloride (AAPH) in a sample is known as the ORAC assay and it was performed following Bermúdez-Oria et al (2019). The reactive oxygen species (ROS) produced diminishes the fluorescence signal generated by the fluorescein. The samples were conveniently diluted with sodium phosphate buffer (10 mM, pH 7.4) and 25 µL of the sample was transferred to a microplate. A blank with 25 µL of phosphate buffer was used, while the standards were filled with 25 µL of trolox solutions at concentrations between 10 and 140 µM. Subsequently, 150 µL of 1µM fluorescein was added to all wells. The plate was incubated at 37 oC for 15 minutes. After this time, 25 µL of AAPH (250 mM) were added to each well to initiate the reaction. In a plate reader (Fluoroskan Ascent™, Thermo Scientific™) measurements were taken every 5 minutes for a period of 90 minutes at an excitation wavelength of 485 nm and an emission wavelength of 538 nm. Final ORAC values were calculated using the regression equation between Trolox concentration and area under the curve (AUC), and expressed as µmoles of Trolox equivalents per mmoles of active molecules.

    Line 202: How the homogeneity of results was tested?

    Principal component analysis was not done, it should be performed.

Response: Statistical analyses were not carried out in the solubilization trials of the three main phenols due to the complexity of all the variants used, such as the three storage liquids, all the varieties of olives, and time. Besides, the objective of the work was not to differentiate varieties but to demonstrate that with the wide spectrum of varieties used and the liquids normally used such as brine and the new tendencies such as the addition of acid allow the obtaining of liquid effluents with high concentration in phenols that allow their obtaining and purification.

Reviewer 2 Report

The manuscript is interesting but actually not suitable for publication on Foods. In fact, the aim of the work is not well defined and the paper is done with two separate activities. The evaluation of HT and DHPG production during olive production is the first activity and the evaluation of the biological activities of these compounds is the second activity. But for tests, purified compounds were used then their origin is not significant. In order to have only one work the use of the not purified extract, it is necessary. Then also the title is not correct. The suggestion is to produce two separate works but it is necessary to amply above all the first work. Special remarks:
1) explain the material used and reported on lines 91-94
2) L97: solution in water ?
3) Table 1: origin and date are not necessary since not used for the data discussion (really the origin could be interesting in order to see if there is an effect of production area on these compounds)
4) L111-L116: explain the extraction method
5) L123: what method was used to define the maturity?
6) As reported on the storage liquids there are olives with diffent maturity: this is not correct then all the results obtained are not correct
7) L126: HPLC-UV or HPLC-DAD?
8) for HPLC analysis was used directly the storage solution? What method was used to purify the solution before analysis?
9) L132: MilliQ and acetonitrile are not reported in M&M
10) L135: what is B?
11) L137: RT not TR
12) L141: not reported in M&M
13) explain well the method used for ORAC analysis
14) without a statistical analysis it is not possible to evaluate results. It is not acceptable in a research paper "it seemed" (L216)
15) why on the graph are reported 24 months but 12 months in the note (L224)?
16) why the analysis was performed every 2 months in the first year and only at the end of the second year?
17) it is better to report the maximum of HT and the real value of DHPG at this moment in order to see the differences in varieties (Fig. 3)
18) L263 DHPG
19) L279 what is x ?
20) what method was used to prepare extract? (L293-298)
21) L391 : Ty has not evaluated then this affirmation is not correct
22) L395: the effect of olive variety and ripeness on phenol production was not showed then this affirmation is not correct
23) since for biological evaluation were used pure compounds and not a real extract the conclusions (L398-402) are not correct

Author Response

Reviewer 2:

The manuscript is interesting but actually not suitable for publication on Foods. In fact, the aim of the work is not well defined and the paper is done with two separate activities. The evaluation of HT and DHPG production during olive production is the first activity and the evaluation of the biological activities of these compounds is the second activity. But for tests, purified compounds were used then their origin is not significant. In order to have only one work the use of the not purified extract, it is necessary. Then also the title is not correct. The suggestion is to produce two separate works but it is necessary to amply above all the first work.

Response: The importance of the source of origin in the extraction and purification of phenols is crucial. On the one hand, the ease of extraction and purification will prevent the phenols from reacting or transforming into possible isomers or adducts that alter their functionality, and on the other hand, the purified phenol still contains 5% of other companions that directly influence its activity. That is why it does not have to be the same a purified phenol from one source or another since that 5% can improve or worsen its activity. In this sense it has been demonstrated that a small percentage less than 10% in DHPG can double the antioxidant activity of HT, when HT has been purified above 90%.

This aspect has been inserted in the introduction by the following sentence: "It is important the type of source used not only by the main phenols but also by their possible companions once they have been purified and also for the fact of making easier their extraction avoiding undesired reactions".

Special remarks:

1) explain the material used and reported on lines 91-94

Response: All the material used has been reported in a new apart “2.2 Chemicals” and explain in each method in which they are used.

2) L97: solution in water ?

 Response: In PBS, it has been changed in the text.

3) Table 1: origin and date are not necessary since not used for the data discussion (really the origin could be interesting in order to see if there is an effect of production area on these compounds)

Response:  Table 1 has not been eliminated so that it can serve as a reference for the values obtained and although it has not been discussed whether it can serve to distinguish varieties by geographical area and the possible implication of water or soil stress effects, for future work.

4) L111-L116: explain the extraction method

Response: In the method it is detailed "Both methods are based on physical chromatographic systems that allow the extrac-tion of natural compounds without any organic solvent or chemical or enzymatic re-actions". It is not possible to extend the information since both systems are being used at industrial level, in addition, the patents can be consulted to carry out the purification for research purposes.

5) L123: what method was used to define the maturity?

Response: It is already referenced LINE 90. The method for the determination of the ripeness is the one described by García et al., 1996. The description of this method has been insert in the text.

6) As reported on the storage liquids there are olives with diffent maturity: this is not correct then all the results obtained are not correct

Response: In the description it was mistakenly put that there were three states of maturity, when in reality only two were studied separately. This has been corrected in the text: "two state of maturity (green and ripe)". Each of the tests was carried out with only one state of maturity, so the samples with the same maturity can be compared with each other, and the tests have been carried out correctly.

7) L126: HPLC-UV or HPLC-DAD?

Response: HPLC-DAD, it has been corrected in the previous apart.

8) for HPLC analysis was used directly the storage solution? What method was used to purify the solution before analysis?

Reference: For the analysis the samples were directly used after a filtration.

9) L132: MilliQ and acetonitrile are not reported in M&M

Response: It has been included.

10) L135: what is B?

Response: B is acetonitrile.

11) L137: RT not TR

Response: It has been corrected.

12) L141: not reported in M&M

Response:

13) explain well the method used for ORAC analysis

Response: IT has been explained inserting the following paragraph:

The inhibition of oxidation induced by peroxyl radicals produced by thermal oxidation of 2,2′-azobis (2-amidine-propane) dihydrochloride (AAPH) in a sample is known as the ORAC assay and it was performed following Bermúdez-Oria et al (2019). The reactive oxygen species (ROS) produced diminishes the fluorescence signal generated by the fluorescein. The samples were conveniently diluted with sodium phosphate buffer (10 mM, pH 7.4) and 25 µL of the sample was transferred to a microplate. A blank with 25 µL of phosphate buffer was used, while the standards were filled with 25 µL of trolox solutions at concentrations between 10 and 140 µM. Subsequently, 150 µL of 1µM fluorescein was added to all wells. The plate was incubated at 37 oC for 15 minutes. After this time, 25 µL of AAPH (250 mM) were added to each well to initiate the reaction. In a plate reader (Fluoroskan Ascent™, Thermo Scientific™) measurements were taken every 5 minutes for a period of 90 minutes at an excitation wavelength of 485 nm and an emission wavelength of 538 nm. Final ORAC values were calculated using the regression equation between Trolox concentration and area under the curve (AUC), and expressed as µmoles of Trolox equivalents per mmoles of active molecules.

14) without a statistical analysis it is not possible to evaluate results. It is not acceptable in a research paper "it seemed" (L216)

Response: The statistics on the solubilization of the main phenols in the three stored liquids used and in the great quantity of varieties throughout two years of study have not been possible to carry out due to the complexity of all the variables and because the objective of the project is not to distinguish the varieties among them but to be able to determine the potential of each one of the liquid effluents as sources for the obtaining of the studied phenols. In this sense, the statement "it seems" has been eliminated from the text and changed to "the trend was".

15) why on the graph are reported 24 months but 12 months in the note (L224)?

Response: It has been corrected.

16) why the analysis was performed every 2 months in the first year and only at the end of the second year?

Response: Initially the study was planned until the concentration of the phenols began to decrease, this occurred at one year. The sampling done at two years was done because it is the average time limit in which the olives remain in these liquids at industrial level.

17) it is better to report the maximum of HT and the real value of DHPG at this moment in order to see the differences in varieties (Fig. 3)

Response: The maximum of HT was represented against the maximum of DHPG because the interesting thing is to look for sources of one or the other indistinctly, since and as it has been exposed in the work, sources that contain a lot of DHPG are looked for at industrial level to be able to obtain it without HT. The very high concentrations shown in the maximum values of DHPG are very striking and will help to improve and reduce the costs of its extraction. By finding sources of this type, the feasibility of extracting these phenols will be improved.

18) L263 DHPG

Response: It has been corrected.

19) L279 what is x ?

Response: The units has been correctly insert.

20) what method was used to prepare extract? (L293-298)

Response: Extractions were carried out by chromatographic systems under patents as showed in material and method section.

21) L391 : Ty has not evaluated then this affirmation is not correct

Response: Ty was also analyzed as shown in Figures 1 and 2, and therefore its concentration allows us to evaluate and conclude that the effluents analyzed are a good source of Ty as well.

22) L395: the effect of olive variety and ripeness on phenol production was not showed then this affirmation is not correct

Response: That is right, “extraction” has been changed by “solubilization”, because the effect of olive variety and ripeness on solubilization of phenols was evaluated.

23) since for biological evaluation were used pure compounds and not a real extract the conclusions (L398-402) are not correct

Response: The compounds used were purified from the storage liquids studied, obtaining extracts with a degree of purity of 95%. This is why extracts have actually been used, with the remaining 5% depending heavily on the nature of the original source and may condition the biological properties of the purified compound either positively or negatively.

Round 2

Reviewer 1 Report

The manuscript can be accepted.